# How the 'plates' of a health system can shift, change and adjust during economic recessions: A qualitative interview study of public and private health providers in Brazil's São Paulo and Maranhão states

**Giuliano Russo**[1]*, **Maria Luiza Levi**[2], **Maria Teresa Seabra Soares de Britto e Alves**[3], **Bruno Luciano Carneiro Alves de Oliveira**[4], **Ruth Helena de Souza Britto Ferreira de Carvalho**[3], **Lucas Salvador Andrietta**[5], **Jonathan Gonçalves Filippon**[1], **Mário César Scheffer**[4]

1 Centre for Global Public Health, Institute for Population Health Studies, Queen Mary University of London, London, United Kingdom, 2 Center for Engineering, Modelling and Applied Social Sciences, Federal University of ABC, Santo André, São Paulo, Brazil, 3 Department of Public Health, Federal University of Maranhão, São Luís, Maranhão, Brazil, 4 Department of Medicine I, Federal University of Maranhão, São Luís, Maranhão, Brazil, 5 Department of Preventive Medicine, Faculty of de Medicine, University of São Paulo, São Paulo, Brazil

* grusso@qmul.ac.uk

## Abstract

### Background

Economic recessions carry an impact on population health and access to care; less is known on how health systems adapt to the conditions brought by a downturn. This particularly matters now that the COVID-19 epidemic is putting health systems under stress. Brazil is one of the world's most affected countries, and its health system was already experiencing the aftermath of the 2015 recession.

### Methods

Between 2018 and 2019 we conducted 46 semi-structured interviews with health practitioners, managers and policy-makers to explore the impact of the 2015 recession on public and private providers in prosperous (São Paulo) and impoverished (Maranhão) states in Brazil. Thematic analysis was employed to identify drivers and consequences of system adaptation and coping strategies. Nvivo software was used to aid data collection and analysis. We followed the Standards for Reporting Qualitative Research to provide an account of the findings.

### Results

We found the concept of 'health sector crisis' to be politically charged among healthcare providers in São Paulo and Maranhão. Contrary to expectations, the public sector was reported to have found ways to compensate for diminishing federal funding, having outsourced services and adopted flexible–if insecure–working arrangements. Following a drop in employment and health plans, private health insurance companies have streamlined their offer, at

**Data Availability Statement:** All relevant data are within the manuscript and its Supporting Information files.

**Funding:** This study received support from the Confap-MRC call for Health Systems Research Networks. Specifically, GR received the award from the Newton Fund/Medical Research Council (UK), Grant Reference MR/R022747/1. TS received the award from: Fundação de Amparo à Pesquisa e ao Desenvolvimento Científico e Tecnológico do Maranhão (FAPEMA-Brazil),COOPI-00709/18. MCS received the award from Fundação de Amparo à Pesquisa do Estado de São Paulo (FAPESP-Brazil), 2017/50356-7. The funders had no role in study design, data collection and analysis, decision to publish, or preparation of the manuscript.

**Competing interests:** The authors have declared that no competing interests exist.

times at the expenses of coverage. Low-cost walk-in clinics were hit hard by the recession, but were also credited for having moved to cater for higher-income customers in Maranhão.

## Conclusions

The 'plates' of a health system may shift and adjust in unexpected ways in response to recessions, and some of these changes might outlast the crisis. As low-income countries enter post-COVID economic recessions, it will be important to monitor the adjustments taking place in health systems, to ensure that past gains in access to care and job security are not eroded.

## Introduction and background

It is well-established in the international literature that economic recessions can severely impact population health by increasing poverty and the burden of the diseases associated with scarcity [1]. Losing the job during a crisis is believed to play a role, as rising unemployment has been linked to increase in suicides [2], deterioration of mental health [3] and overall healthcare utilization [4]. Recent evidence from the Great Depression in Europe [5] shows that economic recessions carry long-lasting effects health systems, as health and social care spending are squeezed to balance budgets at the time when they are needed the most. Many scholars have in fact pointed out that government policy responses would have more profound effects on health systems than the economic recession itself [6], as in the past shrinking funding and rationing measures pushed health systems to the brink of collapse in recession-hit countries like Greece [7], Ireland [8], Spain [9] or Portugal [10].

However, less is known on what happens to health systems during the testing times of an economic recession, and how their key institutions and providers adjust to austerity or react to attempts to re-structure services. The nascent literature on post-conflict and fragile countries [11] suggests that the 'plates' of a health system do not disintegrate under stress [12], but rather shift to occupy the space left by old actors and lack of regulation [13]. These adjustments to the new conditions at times penalize the weakest, and often outlast the crisis [14].

After over a decade of sustained growth, between 2014 and 2016 an economic recession hit Brazil hard, in connection to the slump in the world demand for export commodities and to the country's own political instability [15]. Mortality and morbidity increased in the country during that recession, mostly on the back of rising unemployment [16]. As a policy response, the right-wing government introduced economic measures as part of its programme to reduce public spending (the so-called Constitutional Amendment n. 95—EC95/2016), hitting particularly hard the health sector [17]. The Amendment froze federal expenditures as proportion of revenues for the next 20 years, linking health funding increases to inflation levels only [18]. Some estimates [19] projected a decline of R$415 billion of the health budget over the twenty years over the period; other studies showed how State and municipal funding increased to offset flailing federal transfers in the health sector [20]. Nonetheless, the combined effect of rising unemployment and weakening of social protection programmes were convincingly associated with the country's excess deaths between 2012 and 2017 [16], as well as with increasing child mortality and morbidity, particularly for the non-white population [21].

Brazil's healthcare system comprises a publicly funded Unified Health System (SUS, in its Portuguese acronym), comprehensive and free at the point of delivery, and a privately financed health insurance system (*planos de saúde, or* healthcare plans), providing access to private services to its subscribers. Established in 1990, the SUS is widely credited to have improved the

health of Brazilians by implementing a national primary care programme [22] and making a wide range of services available and affordable to the country's large and diverse population [23]. Nonetheless, after 30 years the SUS is marred by health inequalities across its diverse states, insufficient funding, and unclear private sector-public sector collaborations [24]. In the last decades, SUS policies intended to improve access, have increased number of doctors as well as care structures, but were not able change the concentration of more complex care in a limited number of cities [25, 26].

Social Health Organizations (OSS) are private entities providing health services within SUS. This form of public-private partnership was introduced as part of the New Public Management reforms, with the objective of providing more administrative flexibility for the health sector, and allow the public sector to specialise on functions of control, stewardship and regulation, rather than on direct provision of services [27]. Despite their ubiquity in all Brazil's states, no firm evidence exist of their effectiveness and efficiency in comparison to direct public provision of services [28].

Private funds account for more than half (54%) of total health spending, including out-of-pocket medicines and private insurance premiums [29]. Private providers fill several sectors of healthcare provision, from medical practices to hospitals and laboratories, providing services to both healthcare plans and SUS. Access to such private services is mostly granted through health insurance plans, often connected with employment; 24.2% of Brazil's population owned a private health insurance plan in 2018 [20]. In the last 20 years, low-cost walk-in clinics (*Clínicas Populares*, or People's Clinics) have started to provide out-of-pocket private services, mostly of outpatient nature [30, 31]. These People's Clinics represent an alternative for those who cannot afford a private health plan, aspire at seeing a specialist, but want to avoid SUS' waiting lists [32]. Despite their growth in most of Brazil's capital cities, these low-cost clinics have attracted criticism for adopting a mercantilist approach to provision of care, and for offering poor quality services disconnected from any healthcare reference system [33].

Within the country, São Paulo and Maranhão represent two extremes with regards of economic development and regional health inequalities. With over 48 million people, São Paulo state has one of the country's highest per capita income, and 43% of its population is covered by private health schemes (Table 1). At US$ 361, the state's public health expenditures per capita are also among the highest. Conversely, Maranhão has 7 million people; the per capita income is one third of São Paulo's and public health expenditures per capita is US$243, and only 7% of the population is covered by private health plans [20]. São Paulo has also three times the number of physicians per capita as Maranhão, the majority of which employed in the public health care service.

The recent COVID-19 pandemic has also hit Brazil hard, with the country recording the world's second largest number of infections and deaths [34]. Before the epidemic there were

**Table 1. Selected São Paulo and Maranhão health and spending indicators.**

| Indicator | São Paulo | Maranhão | Brazil |
|---|---|---|---|
| Public health spending per capita (2018 USD)* | 361.01 | 243.07 | 362.2 |
| Proportion of population with private health plan* | 43% | 7% | 24% |
| Total physicians per capita* | 2.81 | 0.87 | 2.18 |
| Physicians formally employed in SUS per 1,000 population (2017) ** | 1.64 | 0.28 | 1.29 |
| Intensive and intermediate SUS beds per capita (2017) ** | 0.16 | 0.11 | 0.14 |

Source:

*Andrietta et al (2020).

** Annual Relation of Social Indicators, National Health Facilities Register.

already signs that Brazil's health system was buckling under the austerity measures introduced in the aftermath of the recession, and regional health disparities were widening between rich and poor states [18]. It is unclear how an already weakened system will face the burden of the epidemic, or what the repercussions will be of the ensuing economic recession.

We set out to explore the changes in provision of public and private services that took place in São Paulo and Maranhão as response to the 2014–2016 economic crisis. Drawing from key informants' perceptions, we aim to uncover how the different components of Brazil's health system–the 'plates'–changed and adjusted in response to the pressures brought in by the recession. It is hoped the results from this study will offer an insight on how health systems may react and cope to the COVID-19 crisis, and form an evidence base for policies to mitigate the effects in Brazil and other low-and middle-income countries.

## Materials and methods

### The methodological approach

We used qualitative semi-structured interviews with key informants from São Paulo and Maranhão states to explore the reactions of public and private health players to the 2015 economic recession. This qualitative investigation is part of a wider study on the effects of the economic crisis on Brazil's health system and medical workforce [35], and the two states were selected in representation of high- and low-income settings in the country.

We drew from the literature on economic recessions, health systems and human resources for health [36], as well as from our project's conceptualization of the links between crises and health systems to identify relevant questions [20]. Our starting hypotheses were that the macro effects of a recession transmit to health sectors via austerity measures, impacts on the demand for services, and changes in the labour market; these are felt unevenly across the different parts of a health system, as they are mediated by pre-existing conditions (poverty and epidemiological profile), as well as role and degree of development of the private sector. The theoretical aspects of our assumptions are published elsewhere [20].

The interview guide comprised sections on: (a) broad challenges for the health system and labour market; (b) the impact on the demand for health services; (c) the key health policies introduced in response to the crisis as well as the so-called austerity measures; (d) the strategies the public and private players adopted to cope with the challenges posed by the recession, and; (e) the impact of the recession on the livelihoods and professions of those interviewed. It was first drafted during a research workshop of the wider project in June 2018, refined through iterative meetings in the following months, and piloted in São Luís do Maranhão in December 2018 (see S1 and S1 Annexes).

The 21 items checklist from the Standards for Reporting Qualitative Research (SRQR) was used to inform the qualitative approach of the research and report on the interview findings [37]. Ethical approvals were obtained from the Research Ethics Committees of the Federal University of Maranhão (CEP UFMA 3.051.875), and of the Faculty of Medicine of the University of São Paulo (CEP FMUSP 3.136.269).

### Data collection

To capture different views of the recession's effects in São Paulo and Maranhão states, a sampling framework was elaborated to identify key informants among health managers, finance officers, physicians and other health workers. These professional profiles were stratified by public and private sector, as well as capital cities Vs hinterland locations (Table 2). Lists of suitable representatives for the two states were drawn for each professional category among policy-makers and health practitioners of the public health system, Social Health Organisations,

**Table 2. List of people interviewed by profession and state.**

| Professional category | States | | | | Total |
|---|---|---|---|---|---|
| | São Paulo | | Maranhão | | |
| | **Public** | **Private** | **Public** | **Private** | |
| Public health manager | 4 | - | 3 | - | 7 |
| Public finance officer | 2 | - | 2 | - | 4 |
| Health manager | 3 | 6 | 4 | 4 | 17 |
| Physicians | 4 | 3 | 3 | 3 | 13 |
| Other health professionals | 2 | - | 2 | 1 | 5 |
| **Total** | **15** | **9** | **14** | **8** | **46** |

private hospitals, private health insurance companies, and People's Clinics in the two states (Table 2). At least three attempts were conducted to contact and set up interviews with the identified informants.

A total of 46 informants were interviewed in São Paulo and Maranhão states between December 2018 and December 2019. The interviews were conducted by two teams of four local researchers, integrated by a UK researcher for the pilot and subsequent interviews in São Luís do Maranhão. The researchers visited the informants in their place of work (hospitals, clinics of offices) upon telephone appointment. All the interviewees were explained the objectives of the research beforehand and signed an informed consent form. The interviews lasted approximately 1 hour (minimum 53 minutes, maximum 1.35 hour).

## Data analysis

All the interviews were recorded, transcribed and analysed with the support of NVivo software version 10 [38]. Content analysis was used to identify emerging themes and categories [39]. A workshop was organised in São Paulo in June 2018 to train the researchers and data collectors in qualitative analysis and on the use of the NVivo software. A preliminary coding tree (or tree map) was drafted drawing from the interview guide and the first interviews, and each team proceeded to add themes and case nodes as they emerged from the interviews in the two states [40] (Fig 1).

The coding tree comprised eight broad nodes, ranging from the interviewee's reflections on the recent health sector's developments at the country and state level, to the demand for health services, and supply-side bottlenecks during the recession. Specific nodes were created to retrieve and analyse the informants' perceptions on the changes and coping strategies adopted by the public and private providers. Quotes were grouped by nodes and sources, leading to the identification of categories, patterns and themes. Findings were triangulated across different categories of informants and sources of information [41]. Deviant cases were examined to test and corroborate specific interpretations [40]. In a number of cases, indigenous concepts were identified in the transcripts and contributed to the definition of codes to investigate. All the interviews were coded in Portuguese; relevant quotes and passages were then translated into English.

## Results

In this section we first present the interviews findings on how the economic crisis is perceived by different actors in Maranhão and São Paulo, followed by the changes and coping strategies observed by our informants in the public and private sectors as a response to the crisis.

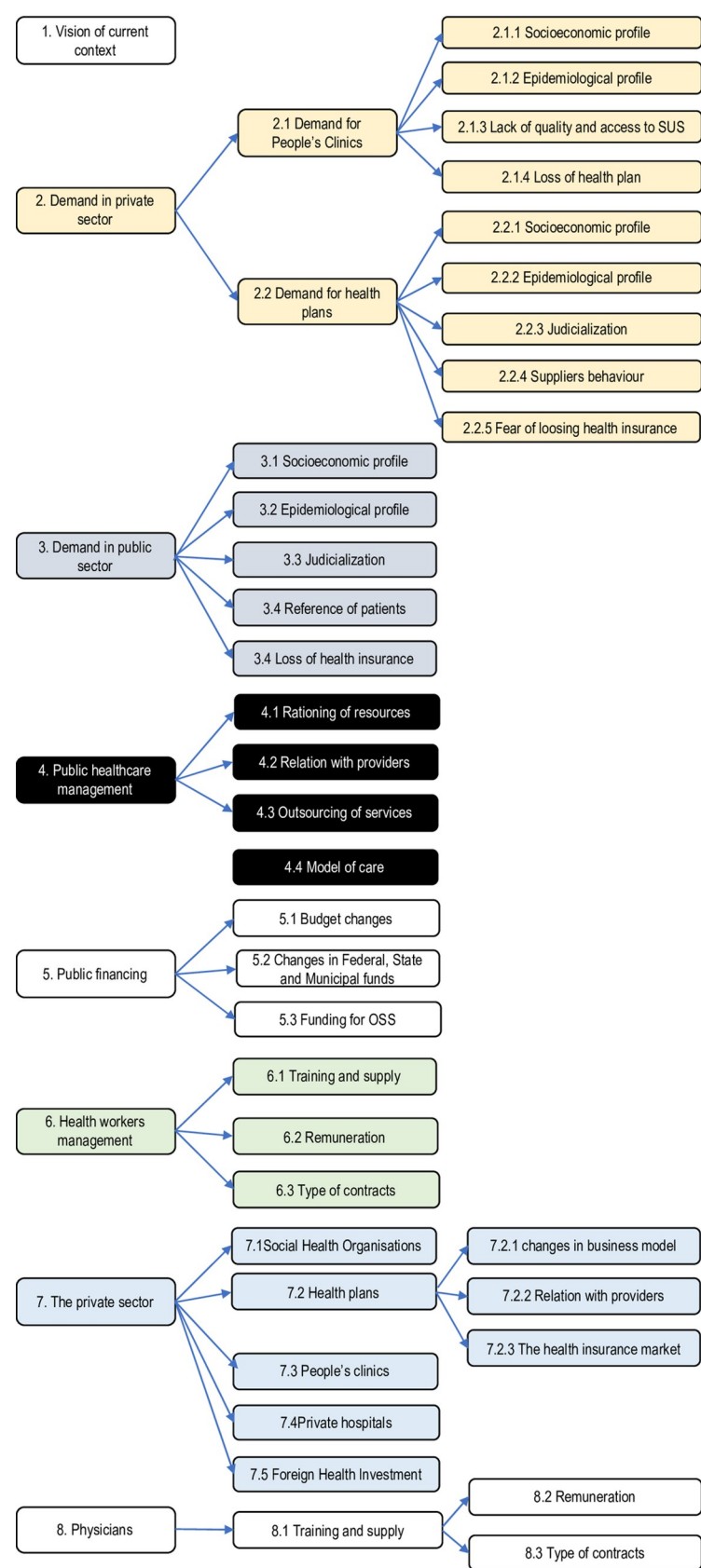

**Fig 1. Coding tree (tree map) for the interview analysis.**

## Conflicting perceptions of crisis

Although it was acknowledged that the country had recently been affected by an economic recession, our informants often struggled to give a specific definition for the term 'crisis': Different perceptions instead emerged of what the recession represented in the two states in relation to: (a) the interpretation of 'crisis' and of its determinants; (b) its reverberations in the health sector, and; (c) its intensity, timing and specific effects for the health sector. We offer below a typology of the interpretations of 'crisis' given by our interviewees.

- Crisis as a predominantly economic phenomenon. Many interviewees identified the word 'crisis' as an economic recession or slowdown, exemplified in terms of the poorest sections of the population "having less money in their pockets", "people losing their employment and being pushed into poverty" or not being able to "afford food, transport or basic health services". Such phenomenon was often linked to the austerity measures adopted by the government during such period, and to their impact on provision of public services, budget cuts, closure of hospitals and expenditure freezes for the public sector.

- Some informants explicitly referred to the SUS crisis, with reference to the pre-existing shortage of resources and political support for the public system. The public healthcare system was also seen as having been unfairly singled out for cuts, and not to enjoy the support of the current government [18, 42]

  *". . .this is how we live here, in perennial crisis! And maybe the health sector is not even being affected as much as other sectors, as this is a social sector, like the education sector; money will always be found for such areas". (Policy-maker from MA-004)*

- Crisis as a political phenomenon. For the last decade, the country and the individual states have gone through years of political instability, with intensification of diverse popular protests with multiple agendas and healthcare demands [43]. After thirteen years of left-wing government, the most recent period represents a marked shift to the right of the political centre, culminated in a shady impeachment process, and in the election of Bolsonaro in 2018 [44]. When prompted with the word 'crisis', some informants directly referred to the consequences of such political instability or to some controversial health financing policies taken these years [45], better known as austerity measures;

- Politicization of the term 'crisis'. Depending on their political affiliation, some of our informants spoke of the current crisis as "inherited from past policies or management". Since the impeachment of Brazil's last president, every aspect of the public life in Brazil is considered to have been politicised and attributed to the shortcomings of either the current right-wing or past left-wing governments–what was called as the "polarization of the Brazilian society" in our interviews. In Maranhão, the word 'crisis' was also associated with the State's old political leadership, in power for the last 30 years, and only recently voted out in the aftermath of multiple scandals [46]. Across the respondents from the public sector, views of crisis consistently focussed on the reduction of public funds for the health sector. For private sector respondents, those from São Paulo consistently displayed an entrepreneurial vision of the latest developments of the Brazilian economy, while those from Maranhão more often referred to their own ability to ride out a bad economic turn.

## The impact of economic recession on SUS

For most of our informants, the impact of the recession was particularly visible on SUS, because of: (a) the cuts in public spending; (b) the impact on the poorer section of the population, traditionally SUS' main customers; (c) the changes in case-mix of patients, now affected by diseases linked to poverty and unemployment; (d) the increasing use of private Social Health Organisations (OSS, in its Brazilian acronym,) reportedly for the flexibility offered to hire and fire, and; (e) the new trend of contracting health personnel in the public service in increasingly casual employment.

There were few doubts for our informants that the federal funding was slashed in both states in connection to the EC95 decree introduced in 2016. However, some managers suggested that decentralised state and municipal funds were used to fill the financing gaps for social sectors such as education and health. Such findings were triangulated across different sources and supported by our analysis of the evolution of health financing between 2010 and 2018 [20].

*"We lost the federal funds, but we managed to compensate–not replace at 100%, but somehow compensate. . . with the local revenues, and I would add with [savings from] better management of public resources" (Public manager MA 001)*

The impact of the recession seemed to have been particularly visible among SUS' traditional patients, the changing case-mix of patients seen in public hospitals, and the prevalence of the poverty-related diseases treated particularly in A&E departments. While malnutrition, intestinal worms, accidental burns from use of open fires, and mental health cases were reported to have been more frequent in SUS facilities, the latter in connection to loss of employment.

*[. . .] so this is like planning the budget for your own household; when you lose your job or get your salary halved because of the crisis, the first thing you cut down is the supplementary care and the (private) health insurance. And this spills on the (financing of) public health, as even those who have health insurance stop seeking private services as they cannot afford (the co-payments), and come and seek care here in the public service, which implies increasing the demand for us. . . (Public Finance Manager, MA 003)*

*"[. . .] the appearance of our waiting room in SUS has changed. These days you get there the people [who can afford] three meals a day. The tone of the complaints to our Ombudsman service has changed; it's visible that these are now written by people with a higher education level". (OSS Public sector manager, São Paulo 001)*

Many interviewees reported that SUS is, in these days, overwhelmed by patients who initially sought outpatient care and medical tests in low-cost private clinics in the private sector, only to be re-directed to public hospitals in case their treatment required hospitalisation or complex tests.

Provision of care in SUS was also reported to have changed substantially to cut costs and increase the role of the private sector during the recession. Outsourcing services to private social health organisations–OSS–for individual hospitals, health trusts, or networks of health facilities for whole geographical areas, seem to have become the norm; these are seen as more flexible and administratively leaner than the traditional public sector management options, relieving public managers of the workload and responsibilities of direct management.

*"[. . .] so the public management units are losing their personnel, and trying to find less staff-intensive ways of managing–they too are outsourcing their own areas of work. Intensive Care*

*Units, A&E, private companies are being used to manage those services too. It is a major trend, not only because of the losses in public sector staff, it is the direct administration model that is being completely overhauled". (Public Finance Manager, São Paulo 002)*

In Maranhão, because of a corruption scandal during a past state administration [46], which involved the local OSS, a different model of public sector management has been adopted. It uses a public enterprise–EMSERH–that enjoys private law status and prerogatives to manage 45 hospitals in the state of Maranhão, while still leaving market space to OSS from other states.

*"So when we were appointed, we came up with a public enterprise from Maranhão, called EMSERH (Maranhão Enterprise for Hospital Services). Such enterprise already controls half the hospital network, which means, in three years we went from 0 to 50%. We believe that in 3–4 years we will no longer have private companies managing public hospitals". (Public-private health manager, MA 003)*

In São Paulo, all the new hospital services have been outsourced since 1998, and their management left to OSS. Public management of health services is becoming rare, and the OSS model is preferred as it allows managers to by-pass cumbersome regulation and budget ceilings when contracting health personnel.

*"Direct public management of clinical units has proved to be unsustainable. Laws such as those of fiscal responsibility, the complexity of Direct Management regulation, the pension contributions, the overall instability . . . all this made Public Direct Management impossible, in my opinion". (OSSC1 Public Manager, São Paulo 001).*

In São Paulo, some of our private sector informants implied that the flexible nature of the OSS model has helped improve efficiency; and that in some circumstances, OSS were still able to make a surplus from public sector contracts by changing the way physicians and nurses are contracted. Some of our OSS informants confirmed that during the recession, their organisations actually managed to expand their business.

*"[. . .] you have now easier ways to review your [public sector] contract, you can now change your workforce, using the PJ [pessoa jurídica] mechanism [hiring employees as sole trader companies]; this helped reduce the cost of labour [. . .]. Our budget increased. We went from 220 [million R$] to 360. We are still in a position to attract further business, but we live under the constant danger of dramatic budget cuts. I am (my company is) still sheltered because our area has little competition. Any government that wants to enter in these areas will have to negotiate with (the providers) [of our districts]". (Public sector Manager, São Paulo 002)*

Following this pattern of contracting-out services, new medical staff are mostly outsourced, losing the benefits of long-term contracts such as paid holidays, bonuses and some pension contributions. Our interviewees informed us that doctors in both states are now hired through third party companies or associations (physicians associations and recruitment agencies providing medical services to public hospitals or to the OSS managing such hospitals. The phenomenon of '*pejotização*'–literally, turning a public employee into a private/civil law entity (*pessoa jurídica*, *PJ*) to be contracted as a company—was credited with saving costs and making hiring and firing easier, by-passing the public sector employment red tape, but also increasing casualisation, making vacancies less secure across the sector.

*"Before, the 'pejotização' [hire individual employees as private companies] was only common for night shifts. From 2016–2017 onwards, we have recorded many issues caused by municipalities that are turning public sector physicians to private law entities, which sounds strange. Because this [professional profile] should be a type of physician with a much stronger link with the community and the public sector, but even these are now been offered PJ contracts".* (Public sector physicians, São Paulo 001)

*"At the beginning there were certain categories of physicians who did not want to be treated as PJ–since the 70s there has always been this institution for physicians like the anaesthetists. But now this phenomenon [pejotização] happens in all areas, even for teams of physicians. You can no longer use different forms of contract!" (Public sector manager, São Paulo 001)*

### The impact on the private health sector

The reduction of the demand for private health insurance plans and the development of low-cost People's Clinics (*Clínicas Populares)*, were the two main impacts reported on the private health sector in the two states. According to some of our informants, the recent economic downturn decreased customers for the health private sector; this was considered to have triggered different coping strategies in the industry, which demonstrated an ability to deal with the crisis or even to take advantage of it. Leading private health insurances and plans were able to sustain their income by increasing prices, even after losing clients [20]. Being pitched to lower-income patients, People's clinics faced diverse scenarios in the studied states; in Maranhão, they could benefit of the demand from people who lost health insurance. But in São Paulo, the People's Clinics' market was reported to have shrunk, as the crisis affected those low-income consumers favoured by the previous 2004–2010 economic growth.

As a result of the recession, unemployment increased from 6.5 to 11.6% in Brazil in 2018; many workers lost health insurance coverage linked to job packages, particularly in São Paulo state. It has been estimated [20] that in the aftermath of the recession, 50.4% of people in São Paulo owned a private health insurance, down from a peak of 57.4% the previous year. In the less prosperous Maranhão, only 7% of the population owns a private health insurance plan, but the drop of subscribers was still noticeable, particularly for its capital city, São Luís. According to our informants, such a drop of demand forced the insurance industry to introduce substantial changes in its business model, integrating its services and operations, reducing the breadth of coverage offered, cutting its cost base, and focusing on less regulated products.

In order to cuts costs, health insurance companies (or '*Planos'*, in Portuguese) appear to have adopted a vertical integration strategy [47], and started to own and run the hospitals and the services health insurance patients were entitled to. Paying health providers less was another example of the strategies to cut the cost base. To this regard, managers reported that doctors were now paid less per medical act, and the contracts signed with insurance companies were mostly for short-term and causal employment, as the *Pejotização* phenomenon seems now to be the favoured form of labour hiring in this area (see Table 3).

In São Paulo, the type of coverage offered, as well as type of customer, was also suggested to have changed to streamline the portfolio, cut costs and drop the least lucrative parts off the market. Coverage was said to have been reduced to guarantee predominantly diagnosis, outpatient services, tests and some in patient care. Co-payments and cheaper healthcare providers have been brought in to reduce costs for the private insurance companies (see Table 3 below). The geographical areas of coverage were reported to have been reduced in São Paulo, with customers forced to seek care only with providers of the Greater São Paulo area, in the attempt of

**Table 3. Strategies and supporting evidence on the changing health insurance market in São Paulo and Maranhão.**

| Commercial strategy | Aim(s) and rationale for introducing changes | Supporting quotes |
|---|---|---|
| Vertical integration and changes in the business model | Business strategies aim at cutting costs and promoting efficiency | "This is one of the ways [setting up primary care clinics to run a successful business]. If you do just that, I do not think you will survive. I have already explained that not everyone benefits from this. There are services that get more expensive if you go to a primary care clinic. Who benefits? So, I think whoever manages to build this structure and put the appropriate patients there and the patients who need to be monitored remotely. . . because we can monitor remotely, that is the point. This is a set of things that have to do with primary care, who you set up directly, that you monitor from a distance". (CP Manager, SP 002) |
| | | ". . . so the health plans companies, when they lose their demand and start searching for diversification, they do like Hapvida did, they start working with their own clinics. This is how they manage to survive". (Owner and manager of CP MA002) |
| | | "I think it is relative (the own network). What is important is that you [can exercise] management over the outcome, the team and the outcome. Whether hired or your own. I do not need a hospital. . . I will give you an example. I return to example [of the market leading company] because it is a situation that I am experiencing directly. We have a product that is in the south of Brazil, in Santa Catarina, which does not exist on its own, it is a hundred per cent contracted. But I do not pay a fee for service, I do not pay for procedures, I pay for the outcome. I pay a fee per capita and then per outcome. So, I pay for a primary care clinic, let us say, R$40 per capita per month for them to take care of that population. That clinic, is not mine, I did not put in a brick, I did not hire anyone, I did not spend a penny of labour resources, and there is someone who does it for me. And I pay for that person to do it. So, it is not own there, and I have, on the other hand, here in São Paulo, my thirty own clinics that I built, I invested in the infrastructure, I put wallpaper, I painted, I put a table, I put a computer. Everything was my cost. So, I have both examples. Neither is right or wrong. Both can be right, each within its own scope and limitations." (OSS Manager, SP 003) |
| Differentiations in the range of plans to attract different clients | Offer of different coverage packages to attract different customers and extract surplus | [Talking about cheap peopl's plans and the possibility of expanding beyond the "ceiling" of 25% of the population] "When people buy health insurance today in Brazil, until recently, we sold the idea that was advertised in the booklet. What did people want? Do I want a thousand doctors, two thousand doctors, three thousand hospitals? You just go to a doctor, hey. How many cardiologists do you need? One, a good one. If this guy is not available, I need another good one. You do not need four hundred cardiologists. So, when we start to change our business model, I am no longer selling the volume of cardiologists, I am selling care for your health needs. I know, you do not know [all those cardiologists]. So, when we start to understand this, and to format a system suitable for these real needs, I think we expanded, yes, maybe [we started to provide for] thirty, thirty-two per cent of the population. What we used to sell in the past, twenty-five per cent was a lot. I think eighteen per cent, twenty per cent was more than I was capable of". (Private plans manager, SP 001) |
| | | "Insisting on a model that pays for the procedure is something that makes us unsustainable. . . our need is an outcome, it does not matter how many procedures you are doing, I need to know if the patient had a stroke and the hospital took fifteen hours to pick it up and treat it and the patient suffered sequelae, [or] if he had three hours of entry and managed to save the patient's life. . . . . We sit down to discuss, for example, what is the protocol for treating stage three breast cancer. . . I do not want to know how much it will cost. . . it is a per capita, it is already included in the cost of oncology. . . How much chemotherapy or medicine will cost, that is your problem. If you buy better, you will earn better.. . . . The fact is that we had a double-digit rising cost trend, eighteen per cent, fourteen per cent, twelve per cent, for three, four years in a row. Models help us to expose this fat". (Private plans manager, SP 001) |
| | | "The [traditional] doctor asks for a volume of totally unnecessary exams without any evidence-based criteria. . . We are doing this [reduction of activities] in some niches with some specialities, with extremely high success. . . These doctors start to be paid this way. . . [it is] a capitation modulated by clinical indicators. . .[. . .]. We reduced [it] by forty per cent. Oncology. I do not pay more for sessions. As they are paid for treatment, the more women are diagnosed with cancer in smaller stages, the greater. . . the margin they will have, because a woman with stage one or two cancer spends less than three and four sessions". (Private plans manager, SP 001) |
| | | "[. . .] it is no longer enough saying: "no, my cost base is too high, and I need to pass this on and charge a higher cost to the health plan company". The health plan company is not going to be able to pay me [such a higher cost], and I need to manage to do the same procedure at a lower cost, so that I can sell it to her [the company]". (Private hospital manager, MA 001) |

(*Continued*)

**Table 3.** (Continued)

| Commercial strategy | Aim(s) and rationale for introducing changes | Supporting quotes |
|---|---|---|
| Customers skimming | Scrapping the least profitable clients from the insurance portfolio (elderlies, high-use customers, chronic disease patients) | "In absolute numbers, our portfolio of services decreased. . . essentially due to the reduction in the number of clients, legal entities.. . . . [the market leading company], in addition to reducing because of the crisis, reduced because it gave up many contracts that were considered loss-making. . . . [the market leading company] of the present,. . . managed much more by accountants than by doctors. . . has a lesser tolerance for these contracts.. . . leveraged by the entry of United. . . the Americans understood that they had to reverse a negative result faster. . . . Most likely the deficit [of contracts] did not come from overuse. . . It came from the normal use of the inadequate price. . . the price adequacy [was] to match what was expected to be used with what had to be paid". (Private plans manager, SP 002) |
| Restrictions in the coverage of plans | Reduction of range of services and providers available | [commenting on Prevent Senior plans, individual marketing, health plans specialized in the elderly]. . . "you can only go to these doctors here. If you need to intern, you will need to go through this place here. If you go to an emergency. . . you buy a stream, you do not buy a network, you do not buy a book, you do not buy unrestricted access". (Private plans manager, SP 001) |
| | | "The private health network is changing strategy to offer access to primary services there, and avoid direct access to the whole range of services. In principle, they are blocking access. . . patients may still go directly to the hospital, but this will end up filtering access for the whole health plan. He [the patient] will have to go thought their own health team, as there are already health plans that has an assessment team for chronic diseases management". (Public and private physician, MA 003) |
| Integrating the costs of judicial sues into regular accounting | Rising prices to cover for perspective lawsuits | "We reserve money for that. This is a point of attention, much more than the direct cost, it is the cost of the brand. This is a cost that we highly value. . . One of the strategies was, of course, to improve the service so that that beneficiary does not go to court. The other thing was to use evidence-based medicine to demonstrate that certain things that were required via court order had to be paid for right away". (Private plans manager, SP 002) |

reducing travelling costs, increase volume of services, and generate economies of scale for specific hospitals. (Table 3).

Individual plans in particular were reported to be now discouraged, as these are less flexible and strictly regulated by the law. Instead, larger, collective plans would be strongly favoured by insurance companies, as these not only offer larger pools of customers, but are also less regulated, giving insurance companies the possibility of offering a more convenient range of services (Table 3).

## Evolution of People's Clinics during the economic recession

People's Clinics (*clínicas populares* in Portuguese) are low-cost, mostly outpatient care services popularised at the beginning of the millennium [30, 31], aimed at providing basic out-of-pocket consultations to lower-middle class patients who want to avoid SUS' long waiting lists. People's clinics are not specific to the recession years, but it was suggested in our interviews that they have critically transformed during the lastneconomic slow-down, and they have now come to represent a key player in the provision of healthcare services in Brazil.

According to our informants, these clinics became popular during Brazil's economic expansion, when the country's lower-middle class (class C) became wealthy enough to afford all sorts of commodities, including private healthcare [48]. Despite being private sector players, such clinics are not connected to any form of health insurance plans, and their business model focuses on outpatient specialist visits, services that do not require an overwhelming initial investment, and for which there seems to be an unmet demand in Brazil.

> "70% of medical act in healthcare is outpatient visit. And outpatient visit is also the cheapest of all acts; but from this consultation [the private clinic] generates a string of parallel incomes: contracts with labs, with X-ray services etc.". (CP manager, São Paulo 001)

*"That's right, I think the problem with People's Clinics is that they do not resolve health issues. I mean, they solve the problem of lack of access to primary care and a few basic tests, but if the patient has any serious problems, they would not know where to send him. That is when they send him back to SUS. This is why I am no fan of those clinics that only offer basic primary care". (Private sector physician, São Paulo 001)*

The clinics occupy the space between SUS patients (mostly lower-class, or class D), and those covered by private health insurance and private hospitals (the upper-middle class B and upper-class A). During the economic recession, because of the drop in disposable income for Class C patients, a squeeze of the demand for People's Clinics' demand was expected. But our informants in Maranhão suggested that although a drop in Class C patients was observed during the recession, such clinics also benefitted from the formerly middle-class patients who lost their employment, and consequently their health insurance plan.

It was suggested that the People's Clinics also benefitted from the increase in the availability of physicians, as newly opened private medical schools arrived at increasing by 13% the national supply of physicians [49]. The vast majority of the physicians in People's Clinics hold a primary job in the public service [50]. Since the clinics do not offer inpatient or surgical care beyond the outpatient visit, it was suggested by one informant in Maranhão that such physicians are able to 'ease through' their private patients into the SUS circuit of reference, therefore enabling them to access more advanced levels of care in the public system.

Despite the similarities, key differences were identified in the evolution of People's Clinics in Maranhão and São Paulo during the crisis, namely in the different market niche in the two states, in the differential effects of the recession, and on the stage of development of the institutions. These clinics were reported to have been around in São Paulo since the 1970es; their market has attracted much competition and can now be considered mature and saturated. While in the North East region of Brazil the clinics developed later [51], and happen to be still in their phase of expansion, with one market leader occupying a large share of the market, and a fringe of smaller competitors. According to our interviewees, the different stage of development implied that in São Paulo new forms of cut-cost competition are beginning to emerge, particularly App-based Consulting Room, where private specialists offer outpatient visits in their own facilities, further eroding the space for organised People's Clinics.

*"Now as a doctor you can advertise your availability in an app, just like AirBnB. Customers select the specialist according to proximity and availability, and go and see them in their own clinic or even at home". CP manager, SP 001*

In Maranhão, Telemedicine was reported to be People's Clinics next frontier of development, as well as agreeing contract with the public sector to relieve some of the pressure on SUS and on the long waiting lists.

*"For me Telemedicina is the next step for us if we want to capture the market in the rural areas. The technology is already there, and I have even started a conversation with a firm to provide the technology [. . .] we know it would not be as good as having a doctor making the diagnosis, but if the alternative is not having any visit at all, I think people may go for it". CP manager, MA 001.*

While the private health insurance market (and respective high-end private clinics that feed off health insurances) is very consolidated in São Paulo State (with over 50% of people covered), this is small and incipient in Maranhão, with a coverage of just 6.8% in 2018. As a

consequence, the space People's Clinics occupy in the richer state is rather narrow, and specifically targeting lower-middle-income (C-class) patients without insurance. Conversely, People's Clinics' market space in Maranhão includes that section of the middle class that lost their health plans during the crisis.

> "[. . .] we were lucky here in Maranhão, because we were able to go after the class B patient who lost his health insurance plan. Class C is still our core business, but these new, richer patients helped off-set the drop of patients in connection to the crisis". CP Manager, MA 001

## Discussion

Our interviews revealed the concept of 'health sector crisis' to be far from consensual, meaning different things to different healthcare providers in São Paulo and Maranhão. Reductions in health funding or chronic scarcity of resources for the National Healthcare Service are considered as parts of a crisis, with the term often charged with political undertones. Despite the cuts in federal funding, the public sector was reported to have found ways to compensate by increasing local state and municipal resources, but also by outsourcing services and adopting flexible–if insecure–working contracts. Within SUS, a marked increase in the use of private companies and casualisation of contracts for health professionals were reported to reduce risks and administrative costs. Following the drop of employment and the associated health plans, private health insurance companies appear to have adapted their business model to streamline their offer of products, at times at the expenses of population coverage. Low-cost walk-in clinics were reported to have been hit hard by the recession in São Paulo, while they were credited to have moved to cater for higher-income customers in Maranhão, partly because of the limited coverage of private health plans in the latter state.

We acknowledge that our study is shaped by local specificities that limit the external validity of the findings. First of all, São Paulo and Maranhão states represent extreme cases in Brazil, with the former sporting a very developed private health sector and abundance of resources, and the second characterised by an underdeveloped network, and a mostly low-income population. Secondly, interviews and content analysis were carried out by different researchers and assistants; despite using the same interview guide and coding tree, this may have introduced unwanted heterogeneity [52] in the investigation in the two states. Finally, our results are based on the views expressed by purposely selected informants; although these were triangulated across interviewees and documental sources, it is still possible these did not fully represent the views of health systems' practitioners in the two states, and may not have capture the entire complexity of the phenomenon under investigation [53].

The three interpretations of crisis identified above were often used interchangeably in the interviews, to refer to the dire circumstances in which the country has found itself for the last decade, as well as to the effects on the health system. There was a sense often that effects and causes of the different crises were overlapping, making it all too difficult to identify what causes what in such muddled context. Noticeably, the word 'crisis' seemed to be charged with political undertones for some of São Paulo's informants, who saw the reference to the country's poor economic performance as an attack to the approach adopted by the current president. This politicization of any debate and polarization of the Brazilian society has been noticed already in the literature [54], and some fear it may put at risk the very existence of democratic institutions. Whilst this finding is consistent with the fluidity in interpretation of 'context' in the health policy analysis literature [55], such blurred interpretations of the phenomenon of

economic crisis make it difficult to find common grounds of analysis of the effects recessions, as well as the search for solutions.

Counterintuitively, it emerged that some parts of the private health sector have adapted and thrived during the 2015–2016 economic recession in Brazil. While this countercyclical effect has been observed before in North American crises [56], articulating a comprehensive explanation for the Brazil's case is not straightforward. In Portugal, the negative impact of austerity measures on the public healthcare was one of the reason why patients started seeking care elsewhere [57]. But there was little evidence in our interviews that Brazil's SUS deteriorated during the latest recession. A possible explanation for the private sector's growth may be that demand for healthcare services does not really decrease during economic recessions, despite the contraction of patients' disposable income. As many workers lost their health plans, such a stable demand for healthcare services would open up spaces for People's Clinics and private hospitals. How the private sector reacts during a recession is particularly relevant for the expected post COVID-19 economic recession, as there are already reports of empty beds and lay-offs in private hospitals [58].

It is doubtful whether the growth of the private sector observed in Brazil during the last recession has truly contributed to expand the offer of quality health services accessible for everyone. Many scholars consider that the private sector is an integral part of a health system [59], and if harnessed properly, it can help expand universal health coverage [60]. However, such notion is heavily contested in many quarters in Brazil [61], as access to healthcare is enshrined in the country's Constitutions as a human right [18], and having to rely on the private sector to provide health coverage would offer a more limited and diminished version of a universal health systems. Our interview accounts on Brazil's People' Clinics and Private Health Insurance plans during recession suggest that the private sector's provision of care would be less than ideal, as: (a) some of the services offered by low-cost clinics would not be integrated with in-patient and higher-complexity care; (b) less wealthy patients may get skimmed off by comprehensive insurance coverage to keep private operations profitable during a recession, and; (c) the rest of the population would be left with no choice but to seek care in crowded and underfunded public facilities. Such findings lend support to the views that in health systems in low- and middle-income countries the growth of the private cannot come at the expenses of the public sector [62], and that a good-quality and accessible public sector will lead to a private health system with these same features [63]. Other social science disciplines and bodies of literature [64] may be needed to better understand performance, functions and interactions of the private and public components of a complex health system during an economic recession.

As the world enters a period of economic uncertainty in the aftermath of the COVID-19 epidemic, it is crucial to ponder what the relevant lessons are from this study in rich and poor states in Brazil. In line with what observed in other countries [10, 65], we show that economic recessions have nuanced impacts on the provision of health services, as the key plates of health systems shift, opportunities open up, and some players are likely to benefit more than others from the new conditions. Despite the squeeze in resources, not all adjustments taking place during a period of crisis will necessarily be for the worse, such as for the increased use of remote technology to expand access to services during this COVID-19 epidemic. Other adjustments may reduce job security and rights for health professionals, as outsourcing services is increasingly used to expand surge capacity during crises, and the individual, small clinics are finding it hard to survive the COVID-19 recession. Many of these adaptations will outlast the crisis, and are likely to have consequences for the sustainability of a national health system. It will be crucial for governments to keep the pulse of health markets, monitor the changes as they take place, and guarantee that workers' rights and population's access to quality services is not penalised by the new equilibria.

## Conclusions

It is well established that economic recessions and associated austerity measures have an impact on health expenditures and population health. Less is known on the consequences for health systems' public and private components, particularly in low- and middle-income countries. This is particularly relevant given the present COVID-19 crisis that will put at risk the health gains of the last decades, particularly for Latin American countries. Brazil has a complex public and private health system, and during the latest economic recession radical measures were taken to balance budgets and improve productivity. Signs are that equilibria between the macro-components of its health system–*the plates*–have shifted in response to recession and associated austerity measures.

As part of a larger research project on recession, health systems and workforce in Brazil, between 2018 and 2019 we conducted semi structured interviews with health practitioners, managers and policy-makers to understand the system effects of the recession in one affluent and one low-income state. We drew from the literature and from the project's own previous conceptualisation to explore the effects the last economic crisis produced on the health systems of the two states, and the adaptation strategies by their health workers. We used the Standards for Reporting Qualitative Research to analyse, triangulate and report our qualitative findings.

We found that the concept or nature of 'health sector crisis' is far from consensual in Brazil, and charged with political undertones. Local state and municipal resources may have compensated for the cuts in public federal transfers, but outsourcing of services and more flexible and insecure working conditions have become the norm in the public sector. It was suggested that private health insurance companies streamlined their business, and low-cost walk-in clinics moved to occupy the space left by the loss of health plans, at times at the expenses of population coverage. We pondered what the consequences are of such changes for the population's access to a quality, universal health system.

It is hoped that our findings will help anticipate the changes that a likely COVID recession may bring to health systems in LMICs and identify policies to mitigate its effects on population access to quality services.

## Supporting information

**S1 Annex. Interview guide in Portuguese.** Questionnaire used for semi structured interviews (original version in Portuguese).
(DOC)

**S2 Annex. Interview guide in English.** Questionnaire used for semi structured interviews (English version).
(DOCX)

## Acknowledgments

Prof. Liberata Coimbra, Judith Pinheiro, Cristiane de Jesus Almeida, Amanda Villalba, and Regimarina Soares Reis for their help in the collection and analysis of the interview data, and participation in the discussions. All those people who made themselves available to be interviewed, and helped piece together the narrative of the crisis in São Paulo and Maranhão.

## Author Contributions

**Conceptualization:** Giuliano Russo, Maria Luiza Levi, Maria Teresa Seabra Soares de Britto e Alves, Bruno Luciano Carneiro Alves de Oliveira, Lucas Salvador Andrietta, Jonathan Gonçalves Filippon, Mário César Scheffer.

**Data curation:** Giuliano Russo, Maria Luiza Levi, Bruno Luciano Carneiro Alves de Oliveira.

**Formal analysis:** Giuliano Russo, Maria Luiza Levi, Ruth Helena de Souza Britto Ferreira de Carvalho, Mário César Scheffer.

**Funding acquisition:** Giuliano Russo, Maria Teresa Seabra Soares de Britto e Alves, Mário César Scheffer.

**Investigation:** Giuliano Russo, Maria Teresa Seabra Soares de Britto e Alves, Lucas Salvador Andrietta, Mário César Scheffer.

**Methodology:** Giuliano Russo, Maria Luiza Levi, Bruno Luciano Carneiro Alves de Oliveira, Jonathan Gonçalves Filippon, Mário César Scheffer.

**Project administration:** Giuliano Russo, Maria Luiza Levi, Maria Teresa Seabra Soares de Britto e Alves, Lucas Salvador Andrietta, Mário César Scheffer.

**Resources:** Mário César Scheffer.

**Software:** Maria Teresa Seabra Soares de Britto e Alves, Bruno Luciano Carneiro Alves de Oliveira.

**Supervision:** Giuliano Russo, Ruth Helena de Souza Britto Ferreira de Carvalho, Mário César Scheffer.

**Validation:** Bruno Luciano Carneiro Alves de Oliveira, Ruth Helena de Souza Britto Ferreira de Carvalho.

**Visualization:** Giuliano Russo.

**Writing – original draft:** Giuliano Russo, Jonathan Gonçalves Filippon.

**Writing – review & editing:** Giuliano Russo, Maria Luiza Levi, Maria Teresa Seabra Soares de Britto e Alves, Bruno Luciano Carneiro Alves de Oliveira, Ruth Helena de Souza Britto Ferreira de Carvalho, Lucas Salvador Andrietta, Jonathan Gonçalves Filippon, Mário César Scheffer.

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
