## [Decision Letter · Decision Letter 0]

3 Sep 2020

PONE-D-20-19510

How the plates of a health system can shift, change and adjust during economic recessions. A qualitative interview study of public and private health providers in Brazil’s São Paulo and Maranhão states

PLOS ONE

Dear Dr. Russo,

Thank you for submitting your manuscript to PLOS ONE. After careful consideration, we feel that it has merit but does not fully meet PLOS ONE’s publication criteria as it currently stands. Therefore, we invite you to submit a revised version of the manuscript that addresses the points raised during the review process.

We look forward to receiving your revised manuscript.

Kind regards,

Jonathan Garcia

Academic Editor

PLOS ONE

Journal Requirements:

2. Please address the following:

- Please please include a copy of the interview guide used in your study in both the original language and English, as Supporting Information.

- Please ensure you have thoroughly discussed all potential limitations of this study within the Discussion section, including the potential introduction of bias during data collection or sampling.

Reviewers' comments:

Reviewer's Responses to Questions

**Comments to the Author**

1. Is the manuscript technically sound, and do the data support the conclusions?

Reviewer #1: Yes

Reviewer #2: Partly

2. Has the statistical analysis been performed appropriately and rigorously? 

Reviewer #1: Yes

Reviewer #2: N/A

3. Have the authors made all data underlying the findings in their manuscript fully available?

Reviewer #1: Yes

Reviewer #2: Yes

4. Is the manuscript presented in an intelligible fashion and written in standard English?

Reviewer #1: Yes

Reviewer #2: Yes

5. Review Comments to the Author

Reviewer #1: Dear colleague,

I found the article quite well developed and very well structured. Compatible with the initial purposes. The topic is very important and deserves, in my view, future development, especially due to the recent events related to the ongoing health crisis in Brazil. Perhaps a longer-term follow-up to a study on the actions of both SUS and the private sector, as well as their interactions.

Best wishes.

Reviewer #2: Considering the economic crisis and recession of 2015 and the subsequent years in Brazil, the article seeks to address, in a qualitative comparative study, the impact of this context on the health care system, on the public and private components. The study is based on data from two Brazilian states, Maranhão and São Paulo, indicating the differences in economic and social development between them.

The main objective of the study is to understand how and if, from the impacts announced above, adequacy, adjustments and changes have occurred in the public and private sectors and what were the results of such impacts on access, coverage and work for the health of the population in the two Brazilian states. The authors point out that the reflection of the article has particular current interest, given that the phenomenon of the pandemic of COVID - 19 has been putting the health care system under stress, considering that Brazil is one of the countries most affected by this pandemic and the Brazilian health care system has already been suffering from the recession of 2015.

It is important for the progress of this report to point out that the reflection on the impact of the COVID-19 pandemic on public and private components of the health care system in Brazil, already threatened by the previous economic recession, was not the central object of the research, conducted prior to the arrival of the disease in the country. However, the authors point out, the reflection from the results of the study can infer questions that support highlights for the COVID-19 relationship and post-pandemic health care systems.

The methodology used was the conduct of 46 semi-structured interviews conducted between 2018 and 2019 with interviews “with health practitioners, managers and policy-makers to explore the impact of the 2015 recession on public and private providers in prosperous (São Paulo) and impoverished (Maranhão) states in Brazil”. I believe there is a balance among the people interviewed, the states and professionals, and also between the public and private components. The authors used NVivo software and followed the Standards for reporting Qualitative Research to provide an account of the findings. I believe that the method was appropriate for the study and for the question presented, although Table 2 "Strategies and quotes on the changing health insurance market in São Paulo and Maranhão” could be better measured and presented.

I consider one of the interesting points of the article the presentation of the different perceptions of the concept of "health sector crisis" among health care provider managers in São Paulo and Maranhão. It is interesting because such a perception may be strongly aligned with the respondents ' answers to the research questions, depending on their affiliation to the public or private sectors and also, how they understand the impacts of the economic recession on the health sector.

In this sense, this point could be better explained for the profile of the interviewees. The profile of the interviewees was pointed out by the authors with a balance between workers of the public and private sectors, managers and professionals, capital and country cities, but the suggestion is that they explain more clearly the performance of the interviewees in the wide public and private health care systems. For example, do the doctors work in clinics, in basic health care services, in hospitals, etc.? I think that a clearer profile of the interviewees gives the reader a broader understanding of the perceptions found in the study.

I suggest that the authors pay close attention to:

1) The Brazilian states represented in the study, São Paulo and Maranhão, are evidently diverse in their economic development process, and the coverage of access to health plans is characterized in this difference, as the authors cite. I suggest that they expand the characterization of the public network - SUS with more data from the two states, considering basic health care services, family health strategy, medium and high complexity, among others. This characterization, even if summarized, can broaden the understanding about the difference in the impacts of economic recession and health in São Paulo and Maranhão.

2) The concepts of systemic complexity and social systems (Niklas Luhmann and other authors) point out that there is a logic of singularities of functionality, construction and interrelation between them. In the case of the central argument of the article, which seeks to understand the changes and impacts of the crisis and recession on the public and private components and their derivations from the health care system, I think that supporting a reflection on the functional and social singularities and the relationship between them is important for the intended analysis. The complexity, in the systemic sense, of the public and private systems, when highlighted, strengthens the reflection on the results.

3) I consider it appropriate to consider the discussion about the covid-19 pandemic in the country and the possible impacts on health policies in Brazil, connecting it to the results of the research. However, I believe that an expansion of the analysis in this respect is necessary. The authors warn, in view of the results of the study, about the fact that some observed changes can last longer than the crisis itself, which is a very important issue to highlight. They also point out that it will be important to monitor the adjustments of the health care system to ensure that previous gains in access to care will not decrease. In this regard, do any signs after 5 months of the arrival of the pandemic in Brazil indicate losses, changes and expansion in the sectors studied?

4) The Proposed Constitutional Amendment (PEC 95) on health financing, among other sectors, has shown harmful results for the SUS and the right to health, and highlighting those impacts in the text seems appropriate.

The article is very interesting and has significant results to discuss policies and the right to health in times of pandemic. The suggestions indicated in the review seek to strengthen the arguments and analyses presented by the authors.

6. PLOS authors have the option to publish the peer review history of their article (what does this mean?). If published, this will include your full peer review and any attached files.

Reviewer #1: No

Reviewer #2: No

---

## [Author Response · Author response to Decision Letter 0]

1 Oct 2020

1. Editor’s further request

Please ensure that you refer to Table 3 in your text as, if accepted, production will need this reference to link the reader to the Table.

Response: This has been done in the paragraphs right before table 3 (see pages 10-11).

2. Editor’s comments

Response: Headings and subheadings have been changed through the manuscript to meet the journal’s style requirements. All changes are highlighted in yellow in text.

2. Please address the following: 

- Please include a copy of the interview guide used in your study in both the original language and English, as Supporting Information.

Response: Copies of the original language and translated interview guide have been uploaded as S1 and S2, respectively

- Please ensure you have thoroughly discussed all potential limitations of this study within the Discussion section, including the potential introduction of bias during data collection or sampling.

Response: We have now added a paragraph in the Discussion’s limitations section addressing the issue of bias and representativeness in our interview sample (Discussion, 2nd par. pag.17.).

Response: Captions have now been introduced for our Supporting Information files S1 Annex and S2 Annex (pag.25).

3. Reviewer #1: 

I found the article quite well developed and very well structured. Compatible with the initial purposes. The topic is very important and deserves, in my view, future development, especially due to the recent events related to the ongoing health crisis in Brazil. Perhaps a longer-term follow-up to a study on the actions of both SUS and the private sector, as well as their interactions.

Response: We thank reviewer 1 for this king comment. Please note this qualitative paper is part of a wider MRC project on the impact of the 2016-2018 economic recession in Brazil, and we are currently in the process of re-positioning its objectives to cover the impact of COVID-19.

4. Reviewer #2:

I believe that the method was appropriate for the study and for the question presented, although Table 2 "Strategies and quotes on the changing health insurance market in São Paulo and Maranhão” could be better measured and presented.

Response: We have now polished the translation and streamlined the topics in Table 2 (pag.12-14).

I consider one of the interesting points of the article the presentation of the different perceptions of the concept of "health sector crisis" among health care provider managers in São Paulo and Maranhão. It is interesting because such a perception may be strongly aligned with the respondents ' answers to the research questions, depending on their affiliation to the public or private sectors and also, how they understand the impacts of the economic recession on the health sector. In this sense, this point could be better explained for the profile of the interviewees. The profile of the interviewees was pointed out by the authors with a balance between workers of the public and private sectors, managers and professionals, capital and country cities, but the suggestion is that they explain more clearly the performance of the interviewees in the wide public and private health care systems. For example, do the doctors work in clinics, in basic health care services, in hospitals, etc.? I think that a clearer profile of the interviewees gives the reader a broader understanding of the perceptions found in the study.

Response: This is a valid point. However, rather than the professional profile of the respondents or their level of service in the health system, it was the public Vs private sector divide, and the SP Vs Maranhão distinction, that appeared to influence our respondents’ perceptions of crisis. We have now added this point in the Conflicting Perceptions of Crisis section (pag.8, first bullet point). 

I suggest that the authors pay close attention to:

1) The Brazilian states represented in the study, São Paulo and Maranhão, are evidently diverse in their economic development process, and the coverage of access to health plans is characterized in this difference, as the authors cite. I suggest that they expand the characterization of the public network - SUS with more data from the two states, considering basic health care services, family health strategy, medium and high complexity, among others. This characterization, even if summarized, can broaden the understanding about the difference in the impacts of economic recession and health in São Paulo and Maranhão.

Response: We have now introduced a table in the Background section with an overview of key health and financing indicators for the two states and for the whole country (Tab. 1, pag. 4).

2) The concepts of systemic complexity and social systems (Niklas Luhmann and other authors) point out that there is a logic of singularities of functionality, construction and interrelation between them. In the case of the central argument of the article, which seeks to understand the changes and impacts of the crisis and recession on the public and private components and their derivations from the health care system, I think that supporting a reflection on the functional and social singularities and the relationship between them is important for the intended analysis. The complexity, in the systemic sense, of the public and private systems, when highlighted, strengthens the reflection on the results.

Response: We accept the argument about the relevance of Luhmann’s work for some of the arguments made in our paper. However, none of the authors has the sociology expertise necessary to apply and develop this kind of analysis. We have nonetheless inserted a reference in the Discussion about this body of work, noting that other disciplines may help understand better the effects of the crisis on the health system, particularly on the articulation and functions performed by its private and public and private components (Discussion, first paragraph, pag.18).

3) I consider it appropriate to consider the discussion about the covid-19 pandemic in the country and the possible impacts on health policies in Brazil, connecting it to the results of the research. However, I believe that an expansion of the analysis in this respect is necessary. The authors warn, in view of the results of the study, about the fact that some observed changes can last longer than the crisis itself, which is a very important issue to highlight. They also point out that it will be important to monitor the adjustments of the health care system to ensure that previous gains in access to care will not decrease. In this regard, do any signs after 5 months of the arrival of the pandemic in Brazil indicate losses, changes and expansion in the sectors studied?

Response: This paper is part of a wider study on the effects of the economic crisis on the health system in Brazil. We have now added a paragraph reflecting on some of our latest findings on the long-lasting effects of COVID-19 on the system (Discussion, last paragraph, pag. 18).

4) The Proposed Constitutional Amendment (PEC 95) on health financing, among other sectors, has shown harmful results for the SUS and the right to health, and highlighting those impacts in the text seems appropriate.

Response: We have now added a paragraph outlining features and consequences of PRC 95 (Introduction and Background, par.3, pag.3).

The article is very interesting and has significant results to discuss policies and the right to health in times of pandemic. The suggestions indicated in the review seek to strengthen the arguments and analyses presented by the authors.

Response: We thank the reviewers for their helpful and insightful comments.

---

## [Editor Report · Decision Letter 1]

7 Oct 2020

How the plates of a health system can shift, change and adjust during economic recessions. A qualitative interview study of public and private health providers in Brazil’s São Paulo and Maranhão states

PONE-D-20-19510R1

Dear Dr. Russo,

We’re pleased to inform you that your manuscript has been judged scientifically suitable for publication and will be formally accepted for publication once it meets all outstanding technical requirements.

Kind regards,

Jonathan Garcia

Academic Editor

PLOS ONE
---

## [Editor Report · Acceptance letter]

14 Oct 2020

PONE-D-20-19510R1 

How the ‘plates’ of a health system can shift, change and adjust during economic recessions. A qualitative interview study of public and private health providers in Brazil’s São Paulo and Maranhão states 

Dear Dr. Russo:

I'm pleased to inform you that your manuscript has been deemed suitable for publication in PLOS ONE. Congratulations! Your manuscript is now with our production department. 

Kind regards, 

on behalf of

Dr. Jonathan Garcia 

Academic Editor

PLOS ONE